# Development and Effects of Leukemia Nursing Simulation Based on Clinical Reasoning

**DOI:** 10.3390/ijerph18084190

**Published:** 2021-04-15

**Authors:** Aeri Jang, Miok Song, Suhyun Kim

**Affiliations:** Department of Nursing, Nambu University, Gwangju 62271, Korea; aerijang@nambu.ac.kr (A.J.); coffeesong@nambu.ac.kr (M.S.)

**Keywords:** high fidelity simulation training, leukemia, clinical reasoning, nursing education

## Abstract

While the effects of simulation education and the importance of the clinical reasoning process have been well-reported, an acute myelocytic leukemia (AML) patient-care simulation program has yet to be formulated exclusively for the clinical experience of students. This study developed and subsequently applied a simulation program based on clinical reasoning for AML to improve the learning outcomes and describe the learning experience for nursing students. Following a mixed-methods framework, the program’s effects on students’ knowledge were quantitatively measured, while their learning experience was qualitatively measured using self-reflection through journal writing. Differences in the pre- and post-scores between the experimental and control groups were statistically significant for theoretical knowledge and clinical performance. In addition, content analysis of both groups’ journals revealed three themes: (1) transformation into a self-directed learner for understanding the clinical situation, (2) increased awareness of clinical reasoning ability, and (3) embodiment of the clinical reasoning process. Standardizing the developed program’s scenarios prompted the participants’ compliance and engagement, and effectively achieved the learning outcomes. This simulation program aided the assessment of nursing intervention’s effectiveness and suggested objective criteria according to clinical reasoning. Similar programs involving other clinical cases, not exclusive to leukemia, should be developed and evaluated.

## 1. Introduction

The Fourth Industrial Revolution is a recent new-generation technological revolution in information and communication [1]. Thus, current nursing education requires a new paradigm to allow various learning experiences for practical application of the students’ clinical practice [2] and keep pace with the rapid technological advancements. One factor that will lead to nursing education change in the Fourth Industrial Revolution era is education using simulation [3,4]. Educational simulations can help nursing students enhance clinical reasoning skills, gain experience through practice, improve self-confidence, and gradually develop their vision for what constitutes excellent care [5]. Thinking like an expert nurse requires a form of engaged moral reasoning informed by generalized knowledge and rational processes and enhanced through expert guidance and coaching [6].

Although various scenarios involving important diseases in which direct clinical practice is necessary, the number of programs remains sorely lacking. Acute myelocytic leukemia (AML) is a disease in which hematopoietic stem cells become malignant cells, proliferate in the bone marrow, spread to the blood, and invade the liver, spine, and lymph glands. As a result, anemia, white blood cell reduction, platelet reduction, and leukocytosis occur [7,8]. Diseases like AML require a simulation operation scenario as their sudden onset nature with poor prognosis can make nursing care and clinical observation difficult for students [7]. Nursing performance should cope with various realistic situations based on understanding and professional knowledge of the physical and psychological complex state of AML patients [8]. Despite the pressing need for AML simulation education, scenarios involving routine and skin cancer patient care, including chemotherapy [9,10,11], and gynecological cancer patient care, are currently prioritized for development and use in nursing education [12,13]. Therefore, if direct education on AML nursing is not provided, nursing college students may work with limited clinical competence and confidence in AML nursing.

A nurse’s ability to perform clinical reasoning results in rational clinical decisions; thus, improving patient care performance and quality [14]. Clinical reasoning is a critical thinking strategy that verifies and analyzes patient-related data, applies a nursing process to solve the patient’s problem, makes a nursing diagnosis, and establishes a nursing plan accordingly [15]. Therefore, a high level of reasoning ability is related to patient well-being and safety [16], and needs to be educated as a core competency of nursing students. The American Association of Colleges of Nursing (AACN) [17] confirms and emphasizes the importance of clinical reasoning in undergraduate education, describing it as an essential competency for integrative problem-solving.

Clinical reasoning competency can be improved through knowledge acquisition via lectures and repeated clinical practice experience [15,18]. However, clinical practice to promote learners’ clinical reasoning capabilities is limited due to the rising level of medical consumers’ rights, changes in the health care environment, and the Covid-19 pandemic [19]. In addition, simulation effectively improves learner confidence, clinical judgment, and decision-making skills that are part of clinical reasoning [5]. Therefore, the proposal for an education method using high-fidelity simulation (HFS) to supplement clinical practice limitations [15,20] is meaningful. 

The study aimed to (1) develop the simulation program based on clinical reasoning for AML; (2) identify the level and differences of nursing students’ self-confidence, theoretical knowledge, and clinical performance in learning (learning outcomes) after simulation application; and (3) describe the simulation program’s learning experience of nursing students after application of simulation. The improvement of learning outcomes was hypothesized based on Jeffries et al.’s simulation model [21].

## 2. Materials and Methods

### 2.1. Study Design and Participants

Following a mixed-methods framework, the clinical reasoning–based simulation program’s effects on students’ knowledge were quantitatively measured, while their learning experience was qualitatively measured. The study’s participants were undergraduate students from a nursing college in South Korea. As AML patients require complex nursing monitoring and management, students’ theoretical understanding of AML must be reinforced with application scenarios for a more holistic approach in caring for such patients.

The researchers verbally assured the participants of the study’s purpose and process. Participants subsequently submitted signed written and informed consent after a full explanation about their participation. Students who had taken hematology classes from their adult nursing theory courses, who had physical and psychological difficulties participating in education-related team activities, declined to be photographed and video recorded, and disagreed with the confidentiality terms, were excluded from the study. The matched pair method of the G Power 3.19.2 program (Heinrich-Heine-University, Dusseldorf, Germany) was used to calculate the number of participants, with an effect size of 0.5, α = 0.05, and power of 0.95 to achieve a total of 45 for the experimental and control groups, respectively. A total of 96 students (48 in both experimental and control groups) were initially selected after considering the exclusion rate. From this number, two students were subsequently excluded due to issues with personal schedules, while three others had insufficient responses. The final number of participants was 91 (45 in the experimental group and 46 in the control group).

### 2.2. Scenario Development Process

The International Association for Clinical Simulation and Learning (INACSL) Standard of Best Practice [22] was used as a basis for designing the simulation operation class and the leukemia patient treatment guideline for the scenario setup. The design also included clinical reasoning, which involves identifying the signs/symptoms (S), etiological factors (E), and the problem (P) [23].

Content validation was conducted via email: an expert panel composed of two clinical registered nurses (hematology/oncology), four nursing educators (two responsible for the simulation and two specialized in adult nursing), and one internist sent their opinion on scenario appropriateness and errors. After a pilot test, the final simulation scenario was confirmed following edits and supplements. Table 1 shows the simulation operating process of the developed scenario. Appendix A provides more details on the scenario.

### 2.3. Measurement

#### 2.3.1. Quantitative Study Tool

For the quantitative analysis, the researchers conducted a survey to examine three factors: self-confidence, theoretical knowledge, and clinical performance. Questions were formulated for each factor in relation to nursing process performance for AML patient care. Two adult nursing educators and two registered nurses validated the questions for each factor in clinical practice. For clinical performance, both students and researchers conducted the evaluation.

The researchers identified seven questions regarding core competency to evaluate self-confidence in terms of nursing process performance for leukemia patients. The self-confidence level was graded from 1 (“Not at all”) to 5 (“Very likely”) for each question. Total scores ranged from 7 to 35, lowest to highest; a higher score indicated a higher nursing process performance for leukemia patients. Cronbach’s α was 0.818 for this factor. 

A 10-item multiple-choice questionnaire evaluated the students’ theoretical knowledge regarding leukemia patient care. Questions with a content validity index (CVI) score of 0.8 or higher were used. Each correct answer received 1 point, while incorrect and missing answers received 0 points. The score distribution was from 0 to 10. Similar to self-confidence, a higher score indicated higher knowledge of the nursing process for leukemia patients. 

Twenty questions for situational activities that the participants performed were identified. For performable clinical activities, 2 points were given for “Well done”, 1 point for “Average”, and 0 for “Poorly done” or “Not done”. The score distribution was from 0 to 40. A higher score indicated a higher nursing process performance for leukemia patients. The Cronbach’s α used was 0.924. 

#### 2.3.2. Qualitative Study Tool

The experimental group documented their experiences through reflection journals, which were collected and used as qualitative data. The reflection journal is a document organized by the learner for reviewing the lesson, which allows the learner to consolidate all learning points and reflections during the learning process and enables the educator to diagnose educational difficulties [24]. The nursing students were asked the following reflection questions:What do you think your team did best in this scenario?Write down any points you think are lacking in this scenario.Write what you learned or felt through this scenario.

### 2.4. Study Procedure

The simulation was conducted in the university’s high-fidelity patient simulation training center, and data were collected from November 27 through 14 December 2018. A one-week gap in the pre- and post-study evaluations of the control and the experimental group were done to prevent diffusion. The detailed study process is shown in Figure 1.

### 2.5. Analysis Methods

SPSS Win 20.0 (SPSS Inc., Chicago, IL, USA) was used in quantitative data analysis, calculating the participants’ basic demographics, frequency, percentage average, and standard deviation for each question. The chi-square test was used to analyze the participants’ basic demographics, prior theoretical knowledge, clinical performance, and self-confidence. Comparisons between the experimental and control groups regarding self-confidence, theoretical knowledge, and clinical performance were performed using a t-test. Cronbach’s α coefficient was used to validate the measurement reliability. 

Experts in content analysis with sufficient understanding of the current study, but not directly involved in program operation, carried out the qualitative data collection. The collected data were subjected to the content analysis method developed by Krippendorff [25]. An investigator read all reflection journals and extracted core ideas and concepts on the students’ learning experiences during the simulation program. The extracted data were subsequently categorized through interconnection and abstraction. To verify the categories’ credibility, the investigators returned to the original data following these categories, reading, and analyzing them as a whole. Two qualitative research experts reviewed the analyzed results for feasibility. 

### 2.6. Training of Research Assistants

To prevent bias in the results, the investigator did not participate in simulation operation and data collection. Four research assistants (one simulation instructor, one simulation operator, and two data collectors) were trained to perform these tasks instead. The simulation instructor had more than seven years of clinical experience and more than two years of simulation operation experience, while the simulation operator possessed more than two years of clinical experience and five years of experience in simulation operation. Both instructor and operator were trained twice for 2 h each training by the investigator on scenarios and research-related matters. In addition, the instructor participated in a >8 h simulation-related conference. Two data collectors had two training sessions for 1 h each on how to fill out the survey and precautions. Moreover, their understanding of each question item was also confirmed. The investigator conducted debriefing at the end of the simulation.

## 3. Results

### 3.1. Homogeneity Verification

No statistically significant difference regarding the basic demographics between the experimental and control groups was found (Table 2). 

### 3.2. The Scenario’s Effect

After pre- and post-study scores analyses, statistically significant differences in self-confidence (*p* < 0.001), theoretical knowledge (*p* = 0.001), and clinical performance (*p* < 0.001) were found in the experimental group. However, those in the control group showed statistically significant differences in self-confidence (*p* = 0.019) and clinical performance (*p* = 0.002). Upon comparing the pre- and post-study scores between the groups, there were statistically significant differences in theoretical knowledge (*p* = 0.014) and clinical performance (*p* = 0.020), but not in self-confidence (Table 3).

### 3.3. Content Analysis of Reflection Journals

As a result of analyzing the reflection journal, three main themes and eight sub-themes were derived. The learning experiences of clinical reasoning–based simulation were shown as a process of transformation to a self-directed learner for understanding the clinical situation, increased awareness of clinical reasoning ability, and embodiment of the clinical reasoning process. Details are shown in Table 4.

## 4. Discussion

This study developed a patient with leukemia–care simulation program based on clinical reasoning for nursing students. Clinical reasoning is a cognitive process that uses critical thinking strategies in clinical situations as well as a problem-solving strategy that identifies and diagnoses actual and potential problems of patients based on patient-related data [26,27]. To meet the complex needs of AML patients, nursing students must demonstrate excellent clinical reasoning and judgment [28].

Specifically, the program was designed to teach theoretical knowledge, and clinical performance related to the subject provided by prior learning and participate in simulation scenarios and debriefings based on clinical reasoning. Because the study design characteristics of such simulations have been seen to play a mediating role in reinforcing learners’ nursing competence [29], the simulations based on clinical reasoning were effective in achieving the learning outcomes in nursing leukemia patients. It also enabled the study participants to focus on problem-solving by identifying the patients’ main complaints and establishing and applying nursing intervention priorities according to the nursing process scenario flow. Moreover, the program aided the participants in acquiring practical application of the clinical reasoning process effectively. 

Unlike simulation scenarios developed in previous studies, the standardized cues allowed participants to avoid deviating from program flow and engaging them in the scenario. However, because there is very little research on simulation programs based on clinical reasoning and/or evaluating their effectiveness regarding standardized cues [30], it is necessary to develop more diverse clinical cases to run the education programs.

Although diseases like AML can lead to swift death due to infection or internal bleeding if no appropriate treatment is applied [7], nursing education exclusively focuses on basic knowledge and techniques regarding cancer patients [31]. Previous simulation programs focused more on primary patient care, such as skin assessment, pain management, chemotherapy [9,10,11]. This study was specifically designed to comprehensively check the patient’s condition by constructing a clinical reasoning–based scenario as well as identifying and diagnosing a patient’s actual and potential problems. The simulation program developed in this study can be suitable for training systematic thinking to effectively cope with clinical situations requiring complex therapeutic approaches, such as AML, in a safe environment. 

Unlike Kweon’s [32] study, there was no statistically significant difference in self-confidence between the experimental and control groups post-operation of the simulation. Self-confidence directly affected nursing performance, leading to clinical performance improvement [33]; yet even if the simulation scenario was developed in accordance with the learners’ level, sufficient understanding was still required to boost confidence. In addition, analysis of the reflection journals showed that the participants’ self-motivation and confidence increased due to the motivation and encouragement of instructors and the stimulation of their curiosity. Therefore, in the process of acquiring clinical reasoning and making it one’s own, it cannot be said that the operation of the clinical reasoning–based nursing simulation was not effective in improving confidence because the self-confidence can be answered in the self-report questionnaire while recognizing one’s shortcomings. Moreover, interventions targeting these factors are thus necessary to enhance the learners’ self-confidence. Sufficient preliminary preparation and positive support of instructors, as well as and sufficient pre-learning and repeated learning situations, should thus be provided to learners. 

The experimental group’s theoretical knowledge showed a statistically significant increase after the simulation, consistent with prior studies’ results [34,35]. As the current study holistically applied the clinical reasoning process to the scenario flow and debriefing, the cognitive learning effect increased. As the experimental group also underwent self-directed learning through pre-simulation videos and hands-on learning, the effects of learning increased more compared to the control. The participants recognized and tried to supplement their lack of knowledge, becoming self-directed learners by recognizing the importance of repeated learning. These results show that simulation-based clinical reasoning is effective not only as an alternative to clinical practices but also for theoretical education. However, further research is necessary to determine the learning effects’ longevity and its effective utilization in clinical practices. 

In the clinical performance evaluation, the participants’ self-evaluation was significantly lower than that of the operator. The participants were also found to be comparing their clinical performance with the nurses’ core competencies in the journal analysis. If they were unable to answer patient queries or assess the patient’s condition accurately, they associated the results with a clinical performance and underappreciated themselves. The simulation operator should, therefore, disclose full information to the participants. Additionally, the facilitator should provide continuous positive feedback and standardized cues to support participants, eliminating the frustration and enabling them to complete the process. The journals also demonstrated how incorporating clinical reasoning enabled the integration of theoretical knowledge and clinical practice. This change appeared as students became aware of the clinical reasoning–based simulation situation.

The learners were initially unable to understand how the clinical reasoning was applied. However, after the current study program, they recognized and perceived nursing assessment, diagnosis, and intervention according to clinical reasoning and evaluation as part of the overall nursing workflow. It shows that the developed simulation program effectively systematized nursing performance and enhanced work efficiency. 

Generalizing the study’s results remain challenging due to the limited study population. While measures were taken to prevent the spread of the study content between groups, it remains difficult to state that the spread effects were deterred entirely. Having used a self-reported measurement tool cannot also completely remove errors due to the respondent’s subjective interpretation. 

Unexplored topics, such as the comparisons in educational effects of the various clinical cases, the utilization degree of the knowledge and clinical performance skills acquired from the simulation, and the evaluation of their effects limit the study. Subsequently, future studies can identify the causes behind the absence of change in self-confidence despite improvement in clinical performance level and determine the reason behind the difference in clinical performance evaluation between facilitator(s) and students. The feasibility of replacing the theoretical curriculum with the simulation can also be researched in future works.

## 5. Conclusions

The current study developed a standardized simulation scenario involving AML and evaluated its effect using quantitative and qualitative data. After the simulation program’s operation, statistically significant effects on the participants’ theoretical knowledge and clinical performance were found. Analysis of the journals showed that the participants experienced transformation, becoming self-directed learners and having increased awareness of the clinical reasoning ability. Incorporating clinical reasoning also allowed them to integrate both theoretical knowledge and clinical practice. Compared to the existing lecture-based method, results show that the developed program would be a valuable tool to enhance knowledge and clinical performance—the objective of clinical education. 

Through this study’s result, simulation education was proposed to be applied to nursing students and clinical nurses. As the program developed was related to medical diagnosis and nursing diagnosis, it would help assess the nursing intervention’s effectiveness and suggest objective criteria according to clinical reasoning. It is proposed to be used in the new nurses’ residency program or as a refresher education program for experienced nurses to cope with complex and diverse medical environments.

## Figures and Tables

**Figure 1 ijerph-18-04190-f001:**
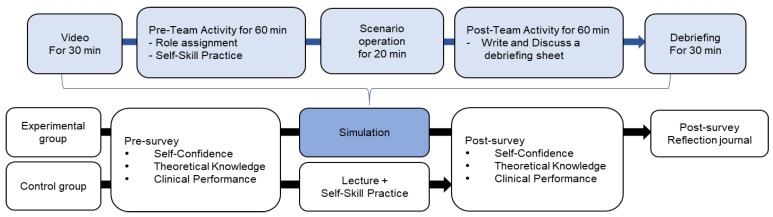
Study process.

**Table 1 ijerph-18-04190-t001:** Simulation set: Continuous bleeding in the mouth after brushing teeth.

Hematological Oncology Internal Medicine Ward			
Process/Monitor Setting	Patient/Simulator Action	Team Activity	Cues/Prompts
**Initial stage** **0–4 min** Mental state: alert. HR ^7^: 68, BP ^3^:100/70, RR ^12^: 24 times/min SpO_2_^14^: 98%, Temp ^15^: 37.6 ° CEKG ^6^: normal, N/S ^10^ 1 L 66 cc/h Oral bleeding, Spot bleeding in both arms and legs Chemo port attached	[**Patient**]: oral bleeding complaints “When I woke up in the morning, I had blood clots in my mouth.” “After brushing my teeth, the bleeding doesn’t stop.” “I have a red mole on my arm.” “My legs look like they have red pepper flakes sprayed on them.” “I don’t have any energy.” “I didn’t have any appetite, so I was unable to eat for several days.”	Self-introduction Identify patient Check: - Oral manifestation - Skin symptoms - Intake/output - Pain - Vitals sign, saturated O_2_ - Anxiety - Bodyweight - Doctor’s prescription - ^9^ IV fluid function - Laboratory check Oral dressing application Report to the doctor about this situation based on SBAR ^13^	No oral manifestations evaluation: “Is it normal to bleed from my mouth like this?” No skin check: “I don’t remember hitting anything, but I noticed contusions.” No intake amount evaluation: “I didn’t have any appetite, so I didn’t eat anything.” No BW ^3^ check: “Is it okay if I didn’t eat anything like this?” No morning lab check: “How was my morning blood work?”
**Clinical Reasoning**	P: Risk for infection. E: Chronic illness S: ANC ^1^, serum WBC, CRP ^5^, Fever, Chemotherapy P: Risk for bleeding E: Treatment regimen S: Oral bleeding complaints, purpura, serum platelets P: Imbalanced nutrition: less than body requirements E: Insufficient dietary intake S: lack of appetite BW ^4^, serum albumin.		
**Action stage** **5–15 min** Mental state: alert HR ^7^: 100, BP ^3^: 80/50, RR ^12^: 32/min, SpO_2_^14^: 96% Temp ^15^: 37.5 °C EKG ^6^: Normal tachycardia Chemo port attached	[**Doctor-post**] “The morning lab have results?”, “If the blood transfusion ordered today is ready, please start right away.” “I changed the antibiotics due to elevated ^5^ CRP. Please check.” [**Event**] During chemotherapy (Adverse effect from chemotherapy) “I feel dizzy.”, “I have a headache and feel nauseous.” (Chills, urticaria, fever, shivering, hypotension) → fever handler activated [**Patient**] Complaining of headache, vomiting, anemia (dizziness) (Important for patients to clarify whether the dizziness started before or after the transfusion started/The current scenario describes the adverse effect of chemotherapy.) [**Doctor**] “How is the saturation?” “Please connect the nasal cannula to O_2_ 2 L/min.”	Application and education of isolation protocol based on ^1^ ANC Normal Saline 1000 mL ^8^ GA education Leg elevation Add doctor’s prescription check If needed, low O_2_ application Chemo port dressing Decitabine start (via chemo port) Metoclopramide injection PC ^11^ collection Transfusion via peripheral ^9^ IV In the event of adverse effects, stop the transfusion Hydrocortisone injection	Doesn’t report to a doctor: “Please call the doctor.” Doesn’t teach saline gargling usage: “The nurse from the earlier shift left this in the morning. What is this?” Doesn’t check the transfusion order correctly: “Why is this transfusion color yellow? Is the color of the blood normally red?” Hangs the blood: “How long does it take to finish?” Reports the side effects during transfusion: “Please slow down the injection and give me the prescribed Hydrocortisone.”
**Final stage** **16–20 min** HR ^7^: 88, BP ^3^:110/70, RR ^12^: 16/min, SpO_2_^14^: 100%, Temp ^15^: 37.5 °C, EKG ^6^: normal Chemo port attached	[**Patient**] “Do I have to eat if I don’t have any appetite?” “Why does the bleeding in the mouth happen?” “Why do I need antibiotics?”	Education AST ^2^ about antibiotic, injection Nursing evaluation Recheck patient condition	Does not teach the mechanisms of leukemia symptoms: “If I’m dizzy and bleeding from the mouth, does that mean I have another disease?” Does not teach post-discharge self-care: “What else do I have to worry about?”
**Debriefing** **30 min**	Ask nursing students the following questions: How do you think you performed patient care as a whole? What is the appropriate clinical reasoning for the scenario situation, and what is the rationale? What are some of the things you learned today that you could use?		

Note. ^1^ absolute neutrophil count, ^2^ antibiotic susceptibility testing, ^3^ blood pressure, ^4^ bodyweight, ^5^ C-reactive protein, ^6^ electrocardiogram, ^7^ heart rate, ^8^ gargling, ^9^ intravenous, ^10^ normal saline, ^11^ platelet concentrates, ^12^ respiratory rate, ^13^ situation, background, assessment, recommendation; ^14^ oxygen saturation as detected by the pulse oximeter, ^15^ temperature.

**Table 2 ijerph-18-04190-t002:** General characteristics and homogeneity of experimental and control groups (*n* = 91).

Characteristics	Categories	Experimental (*n* = 45)	Control (*n* = 46)	χ^2^ or *t*	*p*
		*n* (%) or M ± SD	*n* (%) or M ± SD		
Gender	Female	30 (66.7)	39 (86.7)	4.07	0.053
	Male	15 (33.3)	7 (15.6)		
Age (year)		22.29 ± 2.59	22.89 ± 5.41	−0.68	0.501
Average of prior adult nursing courses		2.93 ± 0.77	2.89 ± 0.71	0.26	0.796
Satisfaction		2.07 ± 0.58	2.24 ± 0.74	−1.24	0.217
Nursing experience with patients with leukemia	Yes	9 (20.0)	5 (11.1)	1.46	0.259
	No	36 (80.0)	41 (91.1)		

**Table 3 ijerph-18-04190-t003:** Mean score comparisons between experimental group and control group per variable (*n* = 91).

Variables	Group	Pre M ± SD	Post M ± SD	*t*	*p*	Post-Pre M ± SD	*t*	*p*
Self-confidence	Exp.	23.07 ± 6.27	28.27 ± 4.58	−4.50	0.000	5.20 ± 7.76	0.79	0.468
	Cont.	23.63 ± 6.82	27.41 ± 9.60	−2.43	0.019	3.78 ± 10.55		
Knowledge	Exp.	4.33 ± 1.83	5.40 ± 1.70	−3.44	0.001	1.07 ± 2.08	2.50	0.014
	Cont.	3.70 ± 1.23	3.67 ± 1.77	0.07	0.944	−0.022 ± 2.07		
Clinical performance	Exp.	16.71 ± 8.73	28.16 ± 6.47	−8.07	0.000	11.44 ± 9.51	2.37	0.020
	Cont.	17.54 ± 7.94	23.46 ± 10.33	−3.21	0.002	5.91 ± 12.49		

**Table 4 ijerph-18-04190-t004:** Students’ simulation learning experience process based on clinical reasoning.

Main Themes	Sub-Themes	Examples
Transformation to a self-directed learner for understanding the clinical situation	Interest in learning and self-motivation	“I finally understand what nurses do in the hospital, so I know now why I need to study.” “I know now how anatomy and physiology I learned were connected to clinical practice, so I am going to review what I don’t know very well.” “I find that I become more familiar with the setting after repetition.”
	Will and commitment to direct their own learning process	“I learned how to wear the mask and gown used in the isolation room.” “The medication calculation was so difficult that I am going to need to learn more accurately for the next time.”
Increased awareness of the clinical reasoning	Realization of the need for the clinical reasoning	“I didn’t realize why we had to take nursing process courses for the clinical reasoning until after the simulation.” “I thought I didn’t need the nursing process based on clinical reasoning because I had been casually writing reports reflecting the nursing process in clinical practices.”
	Recognition of the main learning contents linked to the clinical situation	“I learned the importance of distributing responsibilities,” “I learned how important it is to make the best decision for the patient at each moment.”
	Self-check of core competencies required for clinical reasoning	“We needed to communicate well and discuss what we don’t know, but because we were in a hurry, we lacked in communication.” “I did not assess the patient properly, so I couldn’t carry out the nursing diagnosis or performance accurately.”
Embodying clinical reasoning process	Experience that the quality of nursing is different depending on the nurse’s clinical reasoning ability	“In a clinical setting, I could see the difference in the nursing provided to the patient based on the clinical reasoning ability of a nurse.” “I learned how important a nurse’s role is.”
	Understanding how the process of clinical reasoning was applied in the clinical setting	“I didn’t know how the clinical reasoning was applied in a clinical setting. After the simulation, I learned how a nurse worked according to the clinical reasoning, and I tried to behave similarly.”
	Integration of clinical reasoning into the nursing workflow	“I experienced and now understand the process of how a nurse on a day shift assesses patients objectively and subjectively, diagnoses, and performs interventions during rounds, and evaluated in the afternoon rounds before the day shift ends prior to handing off to the next shift.”

## Data Availability

The data presented in this study are available on request from the corresponding author. The data are not publicly available due to ethical restrictions.

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
