# Peer review of "Development and Effects of Leukemia Nursing Simulation Based on Clinical Reasoning"

_ijerph, 2021, doi:10.3390/ijerph18084190_

Round 1

Reviewer 1 Report

Review

The original article Development and Effects of Leukemia Nursing Simulation based on Clinical Reasoning was aimed on the development of the effects of a clinical reasoning– based on simulation program involving AML patients on nursing students’ learning.

The methodology of the study is adequate. I appreciate using mixed method study in this paper, as it always gives the opportunity to see the problem from few perspectives.

The paper has a potential, but it requires some improvement.

Detailed comments:

  1. Abstract.

 - the aim of the study should be the same in the abstract and in the main text (Abstract, Introduction)

- add the conclusions – Conclusions: Similar programs involving other clinical cases, not exclusive to leukemia, should be developed and evaluated. – this is the recommendation.

- remove numbers before sections

- improve the punctuation

  1. Introduction.

- The explanation of the Acute Myelocytic Leukemia should be added. Short description of the disease.

  1. I also suggest rewriting the "conclusions" section:

-       There is no need to repeat the aim of the study in this section.

-       Limitations of the study and future research directions should be moved to the discussion.

-       Clearly indicate conclusions and recommendations to the practice.

  1. Authors should correct the references section according to instructions for authors.
  2. Acknowledgments. This section should include acknowledgments of people or institutions. It would be worth acknowledging the participants of the study.

Author Response

---------------------------------------------------------------------------------

We would like to thank you and the reviewers for your time and constructive feedback improving our manuscript, “Development and Effects of Leukemia Nursing Simulation based on Clinical Reasoning” (ijerph-1168652), which we submitted to the International Journal of Environmental Research and Public Health. Please see our responses to the reviewers’ comments after this letter.

We hope that the changes we made have significantly improved the quality of our manuscript. Please do let us know if there is anything else we can do at this stage. We would be more than happy to do everything we can to assist you in this process.

We look forward to hearing from you!

Best Regards,

Su Hyun Kim

Reviewer 2 Report

I have some comments/suggestions. 

  1. English translation is not acceptable; there are too many changes to list but it needs significant revision.
  2. Define/discuss 4th industrial revolution - I had to look that up and I would imagine that most readers would need to as well. 
  3. Tenses flip back and forth between past, present and future.
  4. Specify that you are referring to undergraduate/prelicensure nursing students.
  5. AML patients are very ill and require complex nursing monitoring and management; specify the significance of why you have chosen this population to focus on for this simulation. 
  6. Define/discuss clinical reasoning.
  7. Section 2.6 is titled "training of research assistants" yet the focus is on the experience of the investigators but only one sentence is about RAs.
  8. Many abbreviations are used in Table 1 - specify/define. Also, there are multiple instances where it says "you" (the nursing student?). 
  9. Too much information in Table 1 and Figure 2 - these need significant simplification

Author Response

-------------------------------------------------------------------------

We would like to thank you and the reviewers for your time and constructive feedback improving our manuscript, “Development and Effects of Leukemia Nursing Simulation based on Clinical Reasoning” (ijerph-1168652), which we submitted to the International Journal of Environmental Research and Public Health. Please see our responses to the reviewers’ comments after this letter.

We hope that the changes we made have significantly improved the quality of our manuscript. Please do let us know if there is anything else we can do at this stage. We would be more than happy to do everything we can to assist you in this process.

We look forward to hearing from you!

Best Regards,

Su Hyun Kim
